# Universal and Non-Universal Features in the Random Shear Model

**DOI:** 10.3390/e24101350

**Published:** 2022-09-24

**Authors:** Fabio Cecconi, Alessandro Taloni

**Affiliations:** Istituto Sistemi Complessi, Consiglio Nazionale delle Ricerche, Via dei Taurini 19, 00185 Rome, Italy

**Keywords:** universality, anomalous transport, disorder

## Abstract

The stochastic transport of particles in a disordered two-dimensional layered medium, driven by correlated *y*-dependent random velocity fields is usually referred to as *random shear model*. This model exhibits a superdiffusive behavior in the *x* direction ascribable to the statistical properties of the disorder advection field. By introducing layered random amplitude with a power-law discrete spectrum, the analytical expressions for the space and time velocity correlation functions, together with those of the position moments, are derived by means of two distinct averaging procedures. In the case of quenched disorder, the average is performed over an ensemble of uniformly spaced initial conditions: albeit the strong sample-to-sample fluctuations, and universality appears in the time scaling of the even moments. Such universality is exhibited in the scaling of the moments averaged over the disorder configurations. The non-universal scaling form of the no-disorder symmetric or asymmetric advection fields is also derived.

## 1. Introduction

In 1973, Dreizin and Dykhne proposed a model for the density of a diffusion material in stationary current flow, through a disordered conducting medium [1]. In particular, the model was conceived for calculating the effective conductivity of a nonuniform concentration of carriers, describing its stationary transport under the influence of both diffusion and convection (with velocity *U*). The carriers transport was assumed to be two-dimensional and limited to a stripe of transverse dimension *L*, see Figure 1. As the fluctuations of the symmetric conductivity tensor in the transverse direction were taken to be large compared with longitudinal components, the transverse diffusion overwhelms the longitudinal one, i.e., $D\equiv D_{y}\gg D_{x}$. In the transverse (x) direction, the diffusing particles experience random pulsations, which produce narrow convective flow with a velocity *U*. Here, the velocity field has a “quenched” randomness, from which the expression *random shear model* arises. Therefore, the stochastic motion of a generic tracer inside the channel is a combination of two mechanisms. On one side, the tracer undergoes normal diffusion on *y*, the channel’s transverse direction, on the other, it experiences a random static velocity field directed toward the channel’s axis *x*. In terms of Langevin equations for an ensemble of independent particles, the model is hereby defined as
(1)$$\begin{array}{ccc} {\dot{x}}_{i} & = & U(y_{i}) \end{array}$$
(2)$$\begin{array}{ccc} {\dot{y}}_{i} & = & \sqrt{2D}\;\xi_{i}\left(t\right) \end{array}$$
where $\left\{\xi_{i}\left(t\right)\right\}$ are independent, delta-correlated and zero-mean Gaussian thermal noises. Periodic boundary conditions are enforced on the y-direction at y=±L/2, to implement a channel-like geometry along the x-axis.

The above equations were introduced in 1980 by Matheron and de Marsily [2] in their celebrated paper on the solutes transport in porous media. Since its introduction, the most striking feature of the random shear model was its longitudinal superdiffusive behavior, $\langle \delta^{2}x\left(t\right)\rangle ∼t^{3/2}$. Generally speaking, the average 〈⋯〉 is a twofold operation, namely, an average over the thermal noise in (Equation 2) and an average over the ensemble of the disorder layered velocity field. In the late-80s-early-90s period, there was a flourish of mathematical models devoted to the investigation of the Drezin–Dykhne superdiffusive regime and its relation with statistical properties of the quenched advection field. The vast majority of the theoretical and numerical studies assumed the distribution P(U) of the static shear field to be Gaussian, with a stationary autocorrelation function in space, of the type $\langle U\left(y_{1}\right)U\left(y_{2}\right)\rangle ∼\delta \left(y_{1}-y_{2}\right)$ [3,4,5,6,7,8,9,10,11,12]. However, although this simplified picture was largely adopted, the necessary and sufficient conditions for having a superdiffusion $\propto t^{3/2}$ were already clearly stated in the Dreizin–Dykhne paper: i) the stationarity of the advection field autocorrelation function $\langle U\left(y_{1}\right)U\left(y_{2}\right)\rangle =f(|y_{1}-y_{2}\left|\right)$, and ii) its integrability $\int_{-\infty }^{\infty }dy\langle U\left(0\right)U\left(y\right)\rangle <\infty$ [1]. Indeed, Matheron and de Marsily assumed a stationary Gaussian-shaped velocity space autocorrelation, a property which was later used in the analysis reported in References [13,14,15]. However, models releasing either one of the aforementioned hypotheses were also considered. As an example, a stationary although power-law velocity autocorrelation function was proposed in Reference [16] to account for solutes diffusion in certain types of fractured rocks, while the absence of stationarity characterizes the velocity autocorrelation function in Reference [17]. As anticipated by the analysis of Dreizin and Dykhne, any deviation from the stationary short-range velocity autocorrelation function entails a superdiffusive behavior different from the classical $t^{3/2}$ law. In general, the statistical properties of the disordered velocity field are limited to the Eulerian two-point autocorrelation function, with the only exception of the work in [18] which introduced an *n*-point correlation function of the type $\langle U\left(y_{1}\right)U\left(y_{2}\right)\cdots U\left(y_{n}\right)\rangle =\sigma_{n}f_{n}\left(y_{1}-y_{2}\right)f_{n}\left(y_{3}-y_{4}\right)\cdots f_{n}\left(y_{n-1}-y_{n}\right)$, and also substituted the Brownian dynamics along *y*, Equation (Equation 2), with the anomalous fractional diffusion. Furthermore, shear models with a deterministic dependence of the velocity on the coordinate *y* were also considered, together with their impact on the time behavior of the mean square displacement: examples are a linear advecting field $U\left(y\right)=U_{0}y$[19,20,21], a power-law $U\left(y\right)=U_{0}{\left|y\right|}^{\beta }sign\left(y\right)$[22], or a sinusoidal $U\left(y\right)=U_{0}\sin\left(y\right)$ [23].

The entire body of the theoretical works devoted to the random shear model takes into consideration the average of any observable on both the thermal noise and the disorder advection velocity field. Usually, the scaling of the solutes probability density function (PDF) along the channel’s axis *x*, P(x,t), and of its moments, $\langle [x\left(t\right)-x{\left(0\right]}^{m}\rangle$, are studied within this framework [3,5,11]. The interest stems from the typical experimental condition where a stationary flux of water is flowing along a preferential direction, say *x*, with random static orientation, across an ensemble of disordered media, such as several porous samples or fractured rocks, see Figure 1. Imagine now dropping a tracer inside one of these porous samples and observing its dispersion in time. Repeating this transport experiment several times on the same sample mimics the thermal average of the tracer’s diffusion. Performing an additive average over the porous media in the ensemble gives the double average considered in previous works. In Reference [24], we adopted another point of view, which was also partially embraced in Reference [15]. We have calculated the tracers’ PDF in the case of a particular case of symmetric quenched disorder, performing the average over a set of uniformly spaced initial conditions along *y*. This point of view originates from the experimental perspective of having at hand one only sample, namely a single fractured rock or, in general, a single disordered medium, on which the diffusion experiment can be conducted several times. However, in this case a large amount of independent tracers is released inside the channel, a solute density uniformly spread throughout the channel breadth, but at the same position *x*. The diffusion experiment consists in observing how the initial density is dispersed in *x* and in time by the water flux. This corresponds to performing an average over the initial conditions. Repeating this experiment several times yields the statistical or thermal average.

In this paper, we carry out a more thorough and extensive study on both the velocity field statistical properties and the position moments in the random shear model. In doing so, we unveil how the scaling properties exhibited by the velocity field are intimately connected to the spreading in time of the position moments. Generally speaking, the computation of the moments is used to test the scaling form of the propagator P(x,t) in the large-time limit [5,11,24]. In line with the model proposed in Reference [24], we perform our theoretical and numerical analysis by introducing a long-ranged velocity field whose properties can be fully established. We derive a simple and precise scaling framework for moments averaged over the disorder, and for moments averaged over an ensemble of uniformly spaced initial conditions, given a quenched configuration of the advection velocity field. We demonstrate the presence of universal features and lack of universality due to large sample-to-sample fluctuations. Importantly, we analyze and discuss the scaling of the moments in the cases of zero disorder, i.e., for deterministic shear models with long-ranged velocity fields. We show that all the scattered results present in the literature on this subject fit neatly into our mathematical scheme.

## 2. Velocity Field Properties

We discuss the static and dynamical properties of a shear longitudinal field which, according to References [23,24], is generated by a superposition of *M* sinusoidal modes,
(3)$$U\left(y\right)=\sum_{n=1}^{M}{\left|k_{n}\right|}^{\gamma /2}\sin\left(k_{n}y+\phi_{n}\right),$$
or cosinusoidal modes:(4)$$U\left(y\right)=\sum_{n=1}^{M}{\left|k_{n}\right|}^{\gamma /2}\cos\left(k_{n}y+\phi_{n}\right).$$

The wave numbers $k_{n}$ run over the set $k_{n}=2\pi /L\left(1,2,\ldots ,M\right)$ and −1<γ<1. The Equations (Equation 3) and (Equation 4) can be expressed in a compact form as
(5)$$U\left(y\right)=\sum_{n=-M}^{M}\frac{U_{k_{n}}}{2}e^{i(k_{n}y+\phi_{n})},$$
with the following prescriptions:(6)$$\begin{array}{ccc} U_{k_{n}} & = & |k_{n}{|}^{\gamma /2}e^{i\frac{\pi }{2}}=-U_{k_{-n}}\\ k_{n} & = & -k_{-n}\\ \phi_{n} & = & -\phi_{-n} \end{array}$$
holding if Equation (Equation 3) is valid, and
(7)$$\begin{array}{ccc} U_{k_{n}} & = & |k_{n}{|}^{\gamma /2}=U_{k_{-n}}\\ k_{n} & = & -k_{-n}\\ \phi_{n} & = & -\phi_{-n} \end{array}$$
for Equation (Equation 4), with $U_{k_{0}}=0$. This choice carries three important advantages when compared to the previous definitions of the velocity field implemented in former works. First, it is automatically verified that
(8)$$\int_{-L/2}^{L/2}dy\;U\left(y\right)=0.$$

Second, the way the disorder is implemented is through the phases $\phi_{n}$ (n=1⋯,M), each one drawn from a uniform independent distribution on the support [0,2π]. Third, all the statistical properties of the advection shear field can be easily calculated. For instance, the correlation function appears to be
(9)$${\langle U\left(y_{1}\right)U\left(y_{2}\right)\rangle }_{\phi }=\sum_{n=-M}^{M}\frac{U_{k_{n}}^{2}}{4}e^{ik_{n}\left(y_{1}-y_{2}\right)},$$
which yields a discrete spectrum $∼k_{n}^{\gamma }$. Therefore our model recovers the cases studied in Reference [16], where the shear field was introduced through its first (zero) and second (power-law) moments. However, in our formulation we overcome any divergence problem, thanks to the regularization due to the sum over discrete $k_{n}$. Moreover if $U_{k_{n}}$=const, in the limit of M→∞, our formulation recovers the Dikhne model [3,4,5,6,7,8,9,10,11,12].

### 2.1. Static Properties

The four points static correlation function is
(10)$$\begin{array}{cc} {\langle U\left(y_{1}\right)U\left(y_{2}\right)U\left(y_{3}\right)U\left(y_{4}\right)\rangle }_{\phi }= & {\langle U\left(y_{1}\right)U\left(y_{2}\right)\rangle }_{\phi }{\langle U\left(y_{3}\right)U\left(y_{4}\right)\rangle }_{\phi }+{\langle U\left(y_{1}\right)U\left(y_{3}\right)\rangle }_{\phi }{\langle U\left(y_{2}\right)U\left(y_{4}\right)\rangle }_{\phi }+\\ \; & {\langle U\left(y_{1}\right)U\left(y_{4}\right)\rangle }_{\phi }{\langle U\left(y_{2}\right)U\left(y_{3}\right)\rangle }_{\phi }, \end{array}$$
which is different from the expression proposed in [18]. Importantly, Equation (Equation 10) implies ${\langle U{\left(y\right)}^{4}\rangle }_{\phi }=3{\langle U{\left(y\right)}^{2}\rangle }_{\phi }^{2}$, and it can be shown that any even moment enjoys the Gaussian property ${\langle U{\left(y\right)}^{m}\rangle }_{\phi }=\left(m-1\right)!!{\langle U{\left(y\right)}^{2}\rangle }_{\phi }^{m/2}$, while the odd moments vanish. This means that if we look at the probability of having a shear *U* at a given channel’s position *y*, over several disorder configurations $\left\{\phi_{n}\right\}$, i.e., $P_{\phi }\left(U|y\right)$, this would be in the average a Gaussian. In addition, since U(y) is a stationary process in space, its PDF must be the same for any position *y*. This is indeed shown in Figure 2a, where the black curves represent the $P_{\phi }\left(U|y\right)$ calculated at 10 different channel’s positions *y*, and the values of U(y) (Equation 5) are collected over an ensemble of 100 different sets of the quenched phases $\left\{\phi_{n}\right\}$. The fluctuations disappear as long as the number of disorder-quenched configurations increases. Indeed, taking an ensemble of $10^{4}$ different phase arrays $\left\{\phi_{n}\right\}$, one obtains the red curve.

Let us now take a different point of view that, in our opinion, has been overlooked in previous works. Let us consider the velocity field when the disorder is quenched: this corresponds to fixing the values of the phases $\left\{\phi_{n}\right\}$ in (Equation 5), and collect the values of the velocity U(y) for y∈[−L/2,L/2]. The resulting distributions $P_{y}\left(U|\phi \right)$ are shown in Figure 2b (black curves). For large *M*, increasing the sampling of the channel, the fluctuations fade off for almost any random quenched configuration, yielding the Gaussian curve in red. As a matter of fact, the black curves represent the PDF of U(y) for various configurations of the quenched disorder, where *y* are 2000 points uniformly spaced through the channel width, while the red curve is obtained from a denser sampling of $10^{5}$ points. As the variance of both Gaussians depicted in panels (a) and (b) is the same, i.e., $\sigma^{2}=\sum_{n=-M}^{M}\frac{k_{n}^{\gamma }}{4}$, $P_{\phi }\left(U|y\right)=P_{y}\left(U|\phi \right)=P\left(U\right)=e^{-U^{2}/\left(2\sigma^{2}\right)}/\sqrt{2\pi \sigma^{2}}$ and this is true for *almost any* random configuration of the disorder, i.e., any typical $\left\{\phi_{n}\right\}$.

However, it is also true that for specific choices of the disorder, the distribution $P_{y}\left(U|\phi \right)$ may markedly differ from a Gaussian. In Reference [24], we addressed this problem by considering a shear field of the type (Equation 3) and quenched phase configuration $\left\{\phi_{n}\right\}$, where the single phases $\phi_{n}$ could be either 0 or π with probability 1/2. In this paper, we consider a different case, namely the complete absence of disorder, i.e., $\phi_{n}=0$∀n. Specifically, we will consider two situations: an ordered shear field (Equation 3) (the case M=1 has been proposed in [23], as already mentioned), and an ordered shear field (Equation 4). The *y*-dependence of these two fields are reported in the insets of panels (c) and (d) of Figure 2, respectively. It is possible to see that, although the asymmetry of the shear field (Equation 3) U(−y)=−U(y) guarantees the symmetry of $P_{y}\left(U|\phi =0\right)$, on the contrary, the parity of the cosine (U(−y)=U(y)) entails an asymmetric PDF, a situation which was already mentioned in Reference [22] within the context of deterministic power-law shear. Importantly, both the shear field distributions, $P_{y}\left(U|\phi =0\right)$ are not Gaussian, as it can be appreciated in the main panels (c) and (d) of Figure 2.

### 2.2. Dynamical Properties

From the analysis reported above, it appears evident that two types of averages can be performed independently: one is taking a well-defined height *y* on the channel and studying any observable by averaging over the disorder, i.e., over the ensemble of the quenched random phases $\left\{\phi_{n}\right\}$. This is the point of view usually taken in previous works [1,2,5,6,8,9,10,11,12,14,16,22,25,26]. The other point of view is to quench the disorder and subsequently average the observable over its dependence on y∈[−L/2,L/2]. This kind of analysis has been adopted in [24] and partially in [15], and it needs to be handled carefully when taking into account dynamical observables. As a matter of fact, the solution of the stochastic Equation (Equation 2) is $y\left(t\right)=y_{0}+\sqrt{2D}w_{t}$, where $w_{t}$ indicates a Wiener’s process, i.e., $\langle w_{t}\rangle =0$ and $\langle w_{s}w_{t}\rangle =|t-s|$. Hence, the time dependence of the shear flow (Equation 5) is expressed as
(11)$$U\left(t\right)=\sum_{n=-M}^{M}\frac{U_{k_{n}}}{2}e^{ik_{n}\left(y_{0}+\sqrt{2D}w_{t}\right)}\;e^{i\phi_{n}},$$
implying that, for a dynamical observable, the average over *y* coincides with averaging over a uniform set of initial conditions $y_{0}$, i.e.,
$${\langle O\left(t\right)\rangle }_{0}=\frac{1}{L}\int_{-L/2}^{L/2}O\left(y_{0};t\right)dy_{0}.$$

For instance, from (Equation 11) the mean drift of a particle is zero whether the average is taken over the initial conditions or over the disorder, i.e., ${\langle {\langle U\left(t\right)\rangle }_{w}\rangle }_{\phi }={\langle {\langle U\left(t\right)\rangle }_{w}\rangle }_{0}=0$, where ${\langle \ldots \rangle }_{w}$ represents the average over the thermal noise.

While analyzing the two-time correlation function, we first perform the thermal average:(12)$${\langle U\left(t_{1}\right)U\left(t_{2}\right)\rangle }_{w}=\sum_{n_{1},n_{2}=-M}^{M}\frac{U_{k_{n_{1}}}U_{k_{n_{2}}}}{4}e^{i\left(k_{n_{1}}+k_{n_{2}}\right)y_{0}}e^{i(\phi_{n_{1}}+\phi_{n_{2}})}e^{-{\left(k_{n_{1}}+k_{n_{2}}\right)}^{2}Dt_{2}-k_{n_{1}}^{2}D\left(t_{1}-t_{2}\right)}.$$

This expression clarifies how, without any further averaging procedure, the velocity process is neither stationary, nor satisfies the independence on the initial condition and, on top of that, it shows an explicit dependence on the particular choice of the quenched disorder. Now, performing an average over the phases, one easily gets
(13)$${\langle {\langle U\left(t_{1}\right)U\left(t_{2}\right)\rangle }_{w}\rangle }_{\phi }=\sum_{n_{1}=-M}^{M}\frac{U_{k_{n_{1}}}^{2}}{4}e^{-k_{n_{1}}^{2}D\left(t_{1}-t_{2}\right)},$$
for the time-ordered situation in which $t_{1}>t_{2}$. Thanks to the fact that ${\langle e^{i\left(k_{n_{1}}+k_{n_{2}}\right)y_{0}}\rangle }_{0}={\langle e^{i(\phi_{n_{1}}+\phi_{n_{2}})}\rangle }_{\phi }=\delta \left(n_{1}-n_{2}\right)$, the same result is obtained performing the average over the initial conditions, i.e., ${\langle {\langle U\left(t_{1}\right)U\left(t_{2}\right)\rangle }_{w}\rangle }_{\phi }={\langle {\langle U\left(t_{1}\right)U\left(t_{2}\right)\rangle }_{w}\rangle }_{0}$. The equation (Equation 12) is stationary and independent both from the initial conditions and from the specific choice of the disorder taken into account. Moreover, Equation (Equation 12) can also be achieved by applying the following formula [5,9,10,11]
(14)$$\begin{array}{cc} {\langle {\langle U\left(t_{1}\right)U\left(t_{2}\right)\rangle }_{w}\rangle }_{\phi }= & \int_{-L/2}^{L/2}dy_{1}dy_{2}{\langle U\left(y_{1}\right)U\left(y_{2}\right)\rangle }_{\phi }{\langle \delta \left(y_{1}-y\left(t_{1}\right)\right)\delta \left(y_{2}-y\left(t_{2}\right)\right)\rangle }_{w}=\\ \; & \int_{-L/2}^{L/2}dy_{1}dy_{2}{\langle U\left(y_{1}\right)U\left(y_{2}\right)\rangle }_{\phi }P\left(y_{2}-y_{0},t_{2}\right)P\left(y_{1}-y_{2},t_{1}-t_{2}\right) \end{array}$$
where P(y,t) represents the one-dimensional Brownian propagator on a strip of amplitude *L* with periodic boundary conditions, i.e.,
(15)$$P\left(y,t\right)=\frac{1}{L}\left\{1+\sum_{m=-\infty ,m\neq 0}^{\infty }e^{ik_{m}y}e^{-k_{m}^{2}Dt}\right\}$$

Let us now turn to the study of the three (ordered) time velocity correlation functions. The disorder averaged correlation function ${\langle {\langle U\left(t_{1}\right)U\left(t_{2}\right)U\left(t_{3}\right)\rangle }_{w}\rangle }_{\phi }$ is easily shown to be 0. On the contrary, the average over the initial positions yields
(16)$$\begin{array}{cc} {\langle {\langle U\left(t_{1}\right)U\left(t_{2}\right)U\left(t_{3}\right)\rangle }_{w}\rangle }_{0}= & \sum_{n_{1},n_{2},n_{3}=-M}^{M}\frac{U_{k_{n_{1}}}U_{k_{n_{2}}}U_{k_{n_{3}}}}{2^{3}}{\langle e^{i\left(k_{n_{1}}+k_{n_{2}}+k_{n_{3}}\right)y_{0}}\rangle }_{0}e^{i(\phi_{n_{1}}+\phi_{n_{2}}+\phi_{n_{3}})}\\ \; & {\langle e^{i\sqrt{2D}\left(k_{n_{1}}w_{t_{1}}+k_{n_{2}}w_{t_{2}}+k_{n_{3}}w_{t_{3}}\right)}\rangle }_{w}. \end{array}$$

From the properties (Equation 6) and (Equation 7), it appears clear that taking an asymmetric velocity field such as (Equation 3), or its symmetric counterpart (Equation 4), yields in principle different results. After straightforward passages the previous expression becomes
(17)$$\begin{array}{cc} {\langle {\langle U\left(t_{1}\right)U\left(t_{2}\right)U\left(t_{3}\right)\rangle }_{w}\rangle }_{0}= & \sum_{n_{1},n_{2}=-M}^{M}\frac{U_{k_{n_{1}}}U_{k_{n_{2}}}U_{k_{-n_{2}-n_{1}}}}{2^{3}}e^{-k_{n_{1}}^{2}D\left(t_{1}-t_{3}\right)}e^{-k_{n_{2}}D\left(t_{2}-t_{3}\right)\left(k_{n_{2}}+2k_{n_{1}}\right)}\\ \; & e^{i(\phi_{n_{1}}+\phi_{n_{2}}-\phi_{n_{1}+n_{2}})} \end{array}.$$

This correlation function exhibits an apparent dependence on the particular choice of the quenched disorder, which makes this observable unpredictably oscillating around 0. Moreover, since it depends only on the time difference $t_{i}-t_{j}$ ($t_{i}>t_{j}$), it highlights that the shear field (averaged over the initial conditions) is a strict sense stationary (SSS) process [27]. Thanks to this property, the expression (Equation 17), derived under the condition $t_{1}>t_{2}>t_{3}$, is equivalent to those obtained for any other time ordering by a simple permutation of the indices [27,28].

The disorder-averaged four-time correlation function, when $t_{1}>t_{2}>t_{3}>t_{4}$, is defined as
(18)$$\begin{array}{cc} {\langle {\langle U\left(t_{1}\right)U\left(t_{2}\right)U\left(t_{3}\right)U\left(t_{4}\right)\rangle }_{w}\rangle }_{\phi }= & \sum_{n_{1},n_{2},n_{3},n_{4}=-M}^{M}\frac{U_{k_{n_{1}}}U_{k_{n_{2}}}U_{k_{n_{3}}}U_{k_{n_{4}}}}{2^{4}}{\langle e^{i(\phi_{n_{1}}+\phi_{n_{2}}+\phi_{n_{3}}+\phi_{n_{4}})}\rangle }_{\phi }\\ \; & {\langle e^{i\sqrt{2D}\left(k_{n_{1}}w_{t_{1}}+k_{n_{2}}w_{t_{2}}+k_{n_{3}}w_{t_{3}}+k_{n_{4}}w_{t_{4}}\right)}\rangle }_{w}e^{i\left(k_{n_{1}}+k_{n_{2}}+k_{n_{3}}+k_{n_{4}}\right)y_{0}} \end{array},$$
which is equivalent in both cases (Equation 3) and (Equation 4). Thanks to the identity
$${\langle e^{i(\phi_{n_{1}}+\phi_{n_{2}}+\phi_{n_{3}}+\phi_{n_{4}})}\rangle }_{\phi }=\delta \left(n_{1}+n_{2}\right)\delta \left(n_{3}+n_{4}\right)+\delta \left(n_{1}+n_{3}\right)\delta \left(n_{2}+n_{4}\right)+\delta \left(n_{2}+n_{3}\right)\delta \left(n_{1}+n_{4}\right),$$
the final expression reads
(19)$$\begin{array}{cc} {\langle {\langle U\left(t_{1}\right)U\left(t_{2}\right)U\left(t_{3}\right)U\left(t_{4}\right)\rangle }_{w}\rangle }_{\phi }= & \frac{1}{2^{4}}\left\{\sum_{n_{1},n_{3}=-M}^{M}U_{k_{n_{1}}}^{2}U_{k_{n_{3}}}^{2}e^{-k_{n_{1}}^{2}D\left(t_{1}-t_{2}\right)}e^{-k_{n_{3}}^{2}D\left(t_{3}-t_{4}\right)}+\right.\\ & \sum_{n_{1},n_{2}=-M}^{M}U_{k_{n_{1}}}^{2}U_{k_{n_{2}}}^{2}e^{-k_{n_{1}}^{2}D\left(t_{1}-t_{3}\right)}e^{-k_{n_{2}}^{2}D\left(t_{2}-t_{4}\right)}e^{-2k_{n_{1}}k_{n_{2}}D\left(t_{2}-t_{3}\right)}+\\ & \left.\sum_{n_{1},n_{2}=-M}^{M}U_{k_{n_{1}}}^{2}U_{k_{n_{2}}}^{2}e^{-k_{n_{1}}^{2}D\left(t_{1}-t_{4}\right)}e^{-k_{n_{2}}D\left(k_{n_{2}}+2k_{n_{1}}\right)\left(t_{2}-t_{3}\right)}\right\}. \end{array}$$

As in the case of Equation (Equation 14), the same result can be obtained by applying the formula
(20)$$\begin{array}{cc} {\langle {\langle U\left(t_{1}\right)U\left(t_{2}\right)U\left(t_{3}\right)U\left(t_{4}\right)\rangle }_{w}\rangle }_{\phi } & =\int_{-L/2}^{L/2}dy_{1}dy_{2}dy_{3}dy_{4}{\langle U\left(y_{1}\right)U\left(y_{2}\right)U\left(y_{3}\right)U\left(y_{4}\right)\rangle }_{\phi }\\ \; & P\left(y_{4}-y_{0},t_{4}\right)P\left(y_{3}-y_{4},t_{3}-t_{4}\right)P\left(y_{2}-y_{3},t_{2}-t_{3}\right)P\left(y_{1}-y_{2},t_{1}-t_{2}\right) \end{array}$$

Expression (Equation 19) allows drawing the conclusion that the average over the phases entails the disappearance of the $y_{0}$-dependence, a characteristic which is also evident in Equation (Equation 13). Moreover, the explicit dependence on the time difference $t_{i}-t_{j}$ ($t_{i}>t_{j}$) also makes the disorder-averaged velocity a SSS. Hence, ${\langle {\langle U\left(t_{1}\right)U\left(t_{2}\right)U\left(t_{3}\right)U\left(t_{4}\right)\rangle }_{w}\rangle }_{\phi }$ is invariant to the permutations of $t_{1},t_{2},t_{3}$ and $t_{4}$. We now study the four-time correlation function averaged over the initial conditions, i.e.,
(21)$$\begin{array}{cc} {\langle {\langle U\left(t_{1}\right)U\left(t_{2}\right)U\left(t_{3}\right)U\left(t_{4}\right)\rangle }_{w}\rangle }_{0}= & \sum_{n_{1},n_{2},n_{3},n_{4}=-M}^{M}\frac{U_{k_{n_{1}}}U_{k_{n_{2}}}U_{k_{n_{3}}}U_{k_{n_{4}}}}{2^{4}}{\langle e^{i\left(k_{n_{1}}+k_{n_{2}}+k_{n_{3}}+k_{n_{4}}\right)y_{0}}\rangle }_{0}\\ \; & {\langle e^{i\sqrt{2D}\left(k_{n_{1}}w_{t_{1}}+k_{n_{2}}w_{t_{2}}+k_{n_{3}}w_{t_{3}}+k_{n_{4}}w_{t_{4}}\right)}\rangle }_{w}e^{i(\phi_{n_{1}}+\phi_{n_{2}}+\phi_{n_{3}}+\phi_{n_{4}})} \end{array}.$$

A direct calculation yields
(22)$$\begin{array}{cc} {\langle {\langle U\left(t_{1}\right)U\left(t_{2}\right)U\left(t_{3}\right)U\left(t_{4}\right)\rangle }_{w}\rangle }_{0}= & \sum_{n_{1},n_{2},n_{3}=-M}^{M}\frac{U_{k_{n_{1}}}U_{k_{n_{2}}}U_{k_{n_{3}}}U_{k_{-n_{3}-n_{2}-n_{1}}}}{2^{4}}e^{i(\phi_{n_{1}}+\phi_{n_{2}}+\phi_{n_{3}}-\phi_{n_{1}+n_{2}+n_{3}})}\\ \; & e^{-k_{n_{1}}^{2}D\left(t_{1}-t_{2}\right)}e^{-{\left(k_{n_{1}}+k_{n_{2}}\right)}^{2}D\left(t_{2}-t_{3}\right)}e^{-{\left(k_{n_{1}}+k_{n_{2}}+k_{n_{3}}\right)}^{2}D\left(t_{3}-t_{4}\right)} \end{array}.$$

Once again, like in Equation (Equation 17), the explicit dependence on the quenched phase configurations is exhibited.

In view of the above analysis, we can draw the following conclusions about the shear field statistical properties:The shear field is a SSS, both averaged over the disorder and over a uniform distribution of initial conditions${\langle {\langle U\left(t_{1}\right)U\left(t_{2}\right)\cdots U\left(t_{2m+1}\right)\rangle }_{w}\rangle }_{0}\neq 0$ (m≥1), showing an explicit dependence on the quenched disorder $\left\{\phi_{n}\right\}$${\langle {\langle U\left(t_{1}\right)U\left(t_{2}\right)\cdots U\left(t_{2m}\right)\rangle }_{w}\rangle }_{0}\neq {\langle {\langle U\left(t_{1}\right)U\left(t_{2}\right)\cdots U\left(t_{2m}\right)\rangle }_{w}\rangle }_{\phi }\neq 0$ for m>1, with no dependence on $y_{0}$${\langle {\langle U\left(t_{1}\right)U\left(t_{2}\right)\rangle }_{w}\rangle }_{0}={\langle {\langle U\left(t_{1}\right)U\left(t_{2}\right)\rangle }_{w}\rangle }_{\phi }$.

## 3. Position Moments

Owing to the properties investigated in the former section, in the present we show the explicit calculation of the second, third and fourth moment averaged over the phases ${\langle {\langle {\left[x(t)-x(0)\right]}^{m}\rangle }_{w}\rangle }_{\phi }$ and over the initial conditions ${\langle {\langle \left[{x\left(t\right)-x\left(0\right)]}^{m}\right]\rangle }_{w}\rangle }_{0}$. The general formulas that we will adopt are the following [9,10,11,25]
(23)$${\langle {\langle {\left[x(t)-x(0)\right]}^{m}\rangle }_{w}\rangle }_{\phi }=m!\int_{0}^{t}dt_{1}\int_{0}^{t_{1}}dt_{2}\cdots \int_{0}^{t_{m-1}}dt_{m}{\langle {\langle U\left(t_{1}\right)U\left(t_{2}\right)\cdots U\left(t_{m}\right)\rangle }_{w}\rangle }_{\phi }$$
(24)$${\langle {\langle {\left[x(t)-x(0)\right]}^{m}\rangle }_{w}\rangle }_{0}=m!\int_{0}^{t}dt_{1}\int_{0}^{t_{1}}dt_{2}\cdots \int_{0}^{t_{m-1}}dt_{m}{\langle {\langle U\left(t_{1}\right)U\left(t_{2}\right)\cdots U\left(t_{m}\right)\rangle }_{w}\rangle }_{0}$$

### 3.1. Second Moment

While the drift (m=1) is invariably 0, the second moment is the only moment that has the same expression if averaged over the disorder or over a uniform distribution of initial conditions. This is true for any configuration of the phases, including the no-disorder cases, as well as for both choices (Equation 3) and (Equation 4). It can be easily deduced from the definitions (Equation 23) and (Equation 24) by using the correlation function (Equation 13). Hence the time behavior of
$${\langle {\langle {\left[x(t)-x(0)\right]}^{2}\rangle }_{w}\rangle }_{0}={\langle {\langle {\left[x(t)-x(0)\right]}^{2}\rangle }_{w}\rangle }_{\phi }=\sum_{n_{1}=-M}^{M}U_{k_{n_{1}}}^{2}\int_{0}^{t}dt_{1}\int_{0}^{t_{1}}dt_{2}e^{-k_{n_{1}}^{2}D\left(t_{1}-t_{2}\right)}$$
can be studied through the Laplace transform
(25)$$\sum_{n_{1}=-M}^{M}\frac{|k_{n_{1}}{|}^{\gamma }}{s^{2}\left(s+k_{n_{1}}^{2}D\right)}.$$

The pole at 0 dictates the long time behavior ∼Lt, while for short and intermediate times $t\ll \frac{L^{2}}{D}$ the pole $-k_{n_{1}}^{2}D$ dominates. The correct analysis can be performed by inverting (Equation 25)
(26)$${\langle {\langle {\left[x(t)-x(0)\right]}^{2}\rangle }_{w}\rangle }_{0,\phi }=\sum_{n_{1}=-M}^{M}\frac{|k_{n_{1}}{|}^{\gamma -2}}{D}t-\frac{|k_{n_{1}}{|}^{\gamma -4}}{D^{2}}\left(1-e^{-k_{n_{1}}^{2}Dt}\right),$$
and is reported in Reference [24]. We hereby present only the scaling behaviors (see Figure 3):(27)$${\langle {\langle {\left[x(t)-x(0)\right]}^{2}\rangle }_{w}\rangle }_{0,\phi }∼\left\{\begin{array}{cc} Lt^{2} & t\ll \frac{L^{2}}{M^{2}D}\\ Lt^{\frac{3-\gamma }{2}} & \frac{L^{2}}{M^{2}D}\ll t\ll \frac{L^{2}}{D}\\ Lt & t\gg \frac{L^{2}}{D} \end{array}\right.$$

One sees that the asymptotic normal behavior is restored thanks to the presence of the channel’s boundaries, which impose an exponential cut-off to the velocity autocorrelation function (Equation 13) [24]. As anticipated, the unique feature of the mean square displacement (Equation 26) is that ${\langle {\langle {\left[x(t)-x(0)\right]}^{2}\rangle }_{w}\rangle }_{0}$ has the same expression for whatsoever disorder quenched configuration, as it is shown in Figure 3 (blue symbols), and this corresponds exactly to the disorder-averaged MSD ${\langle {\langle {\left[x(t)-x(0)\right]}^{2}\rangle }_{w}\rangle }_{\phi }$ (solid lines).

### 3.2. Third Moment

The position third moment is identically zero in the case of disorder average. On the other hand, it is non-zero if the average over the initial conditions is performed, i.e.,
$${\langle {\langle {\left[x(t)-x(0)\right]}^{3}\rangle }_{w}\rangle }_{0}=3!\int_{0}^{t}dt_{1}\int_{0}^{t_{1}}dt_{2}\int_{0}^{t_{2}}dt_{3}{\langle {\langle U\left(t_{1}\right)U\left(t_{2}\right)U\left(t_{3}\right)\rangle }_{w}\rangle }_{0},$$
where the velocity correlation function is given by (Equation 17). Tracing the same procedure of the second moment, we first calculate its Laplace transform
(28)$$\frac{3!}{2^{3}}\left\{\sum_{\begin{array}{c} n_{1},n_{2}=-M\\ n_{1}\neq -n_{2}/2 \end{array}}^{M}\frac{U_{k_{n_{1}}}U_{k_{n_{2}}}U_{k_{-n_{2}-n_{1}}}e^{i(\phi_{n_{1}}+\phi_{n_{2}}-\phi_{n_{1}+n_{2}})}}{s^{2}\left(s+k_{n_{1}}^{2}D\right)\left[s+{\left(k_{n_{1}}+k_{n_{2}}\right)}^{2}D\right]}+\sum_{n_{1}=-M/2}^{M/2}\frac{U_{k_{n_{1}}}^{2}U_{-k_{2n_{1}}}e^{i(2\phi_{n_{1}}-\phi_{2n_{1}})}}{s^{2}{\left(s+k_{n_{1}}^{2}D\right)}^{2}}\right\}.$$

At first glance, one sees that the third moment carries an explicit dependence on the disorder, in analogy to (Equation 17). Hence, since a generic disorder does not preserve the symmetry x→−x, the third moment cannot be neglected. As a matter of fact, in Figure 4a the time evolution of the third moment is reported for several choices of the quenched disorder $\left\{\phi_{n}\right\}$ (brown solid lines). It appears that a specific trend cannot inferred from the fluctuating behavior of ${\langle {\langle {\left[x(t)-x(0)\right]}^{3}\rangle }_{w}\rangle }_{0}$.

A well-defined non-zero scaling in time appears manifest only by considering the symmetric shear case with zero disorder, namely the field (Equation 4) with $\left\{\phi_{n}\right\}=0$, ∀n. A rather lengthy calculation, briefly reported in the Appendix A, shows that the three scaling regimes of the third moment can be identified on the score of what we have conducted for the second moment. We hereby report the overall result
(29)$${\langle {\langle {\left[x(t)-x(0)\right]}^{3}\rangle }_{w}\rangle }_{0}∼\left\{\begin{array}{cc} L^{2}t^{3} & t\ll \frac{L^{2}}{M^{2}D}\\ L^{2}t^{2-\frac{3\gamma }{4}} & \frac{L^{2}}{M^{2}D}\ll t\ll \frac{L^{2}}{D}\\ L^{2}t & t\gg \frac{L^{2}}{D} \end{array}\right.$$

### 3.3. Fourth Moment

We analyze the case of the fourth moment, starting with the disorder-averaged case
(30)$${\langle {\langle {\left[x(t)-x(0)\right]}^{4}\rangle }_{w}\rangle }_{\phi }=4!\int_{0}^{t}dt_{1}\int_{0}^{t_{1}}dt_{2}\int_{0}^{t_{2}}dt_{3}\int_{0}^{t_{3}}dt_{4}{\langle {\langle U\left(t_{1}\right)U\left(t_{2}\right)U\left(t_{3}\right)U\left(t_{4}\right)\rangle }_{w}\rangle }_{\phi },$$
where the expression (Equation 19) must be inserted. Performing the Laplace transform of the former equation consists of transforming each of the terms appearing in (Equation 19). In the Appendix A, we detail the procedure to highlight the time scaling expression of ${\langle {\langle {\left[x(t)-x(0)\right]}^{4}\rangle }_{w}\rangle }_{\phi }$, which retraces that of the second and third moments reported in the previous sections. We sum up the result of our analysis in the following formula
(31)$${\langle {\langle {\left[x(t)-x(0)\right]}^{4}\rangle }_{w}\rangle }_{\phi }∼\left\{\begin{array}{cc} L^{2}t^{4} & t\ll \frac{L^{2}}{M^{2}D}\\ L^{2}t^{3-\gamma } & \frac{L^{2}}{M^{2}D}\ll t\ll \frac{L^{2}}{D}\\ L^{2}t^{2} & t\gg \frac{L^{2}}{D} \end{array}\right.$$

The analysis of the fourth moment averaged over the initial conditions is more involved as it stems from the formula
$${\langle {\langle {\left[x(t)-x(0)\right]}^{4}\rangle }_{w}\rangle }_{0}=4!\int_{0}^{t}dt_{1}\int_{0}^{t_{1}}dt_{2}\int_{0}^{t_{2}}dt_{3}\int_{0}^{t_{3}}dt_{4}{\langle {\langle U\left(t_{1}\right)U\left(t_{2}\right)U\left(t_{3}\right)U\left(t_{4}\right)\rangle }_{w}\rangle }_{0}.$$

A close inspection of the velocity correlation function (Equation 22) reveals that when the conditions $n_{1}=-n_{2}$, $n_{1}=-n_{3}$ and $n_{2}=-n_{3}$ are satisfied, ${\langle {\langle U\left(t_{1}\right)U\left(t_{2}\right)U\left(t_{3}\right)U\left(t_{4}\right)\rangle }_{w}\rangle }_{0}\equiv {\langle {\langle U\left(t_{1}\right)U\left(t_{2}\right)U\left(t_{3}\right)U\left(t_{4}\right)\rangle }_{w}\rangle }_{\phi }$. Therefore, we can split the fourth moment, averaged over the initial conditions, into two overall contributions:(32)$${\langle {\langle {\left[x(t)-x(0)\right]}^{4}\rangle }_{w}\rangle }_{0}={\langle {\langle {\left[x(t)-x(0)\right]}^{4}\rangle }_{w}\rangle }_{\phi }+\sum_{\begin{array}{c} n_{1},n_{2},n_{3}=-M\\ n_{1}\neq -n_{2}\\ n_{1}\neq -n_{3}\\ n_{2}\neq -n_{3} \end{array}}f\left(\phi_{n_{1}},\phi_{n_{2}},\phi_{n_{3}},\phi_{n_{1}-n_{2}-n_{3}};t\right).$$

The last term represents the explicit dependence of ${\langle {\langle {\left[x(t)-x(0)\right]}^{4}\rangle }_{w}\rangle }_{0}$ upon the quenched configuration of the disorder. For a generic random choice of $\left\{\phi_{n}\right\}$ and large *M* we expect that this term does not give a substantial contribution to the scaling behavior of (Equation 32). Indeed, in Figure 4b, several curves pertaining to random configurations of $\left\{\phi_{n}\right\}$ exhibit small fluctuations around the average value represented by ${\langle {\langle {\left[x(t)-x(0)\right]}^{4}\rangle }_{w}\rangle }_{\phi }$ (black dashed line). Thus, we can conclude that ${\langle {\langle {\left[x(t)-x(0)\right]}^{4}\rangle }_{w}\rangle }_{0}\equiv {\langle {\langle {\left[x(t)-x(0)\right]}^{4}\rangle }_{w}\rangle }_{\phi }$ (Equation (Equation 31)) for almost any random typical disorder.

On the other hand, if the disorder is quenched to zero, namely if the velocity advection field is (Equation 3) or (Equation 4) with $\left\{\phi_{n}\right\}=0$∀n, the second term in the Equation (Equation 32) plays a significant role, leading to scaling forms different than those reported in (Equation 31). These are
(33)$${\langle {\langle {\left[x(t)-x(0)\right]}^{4}\rangle }_{w}\rangle }_{0}∼\left\{\begin{array}{cc} L^{3}t^{4} & t\ll \frac{L^{2}}{M^{2}D}\\ L^{3}t^{3-\frac{5\gamma }{4}} & \frac{L^{2}}{M^{2}D}\ll t\ll \frac{L^{2}}{D}\\ L^{3}t^{2} & t\gg \frac{L^{2}}{D} \end{array}\right..$$

Appendix A reports the calculations needed to derive the expression (Equation 33).

## 4. General Scaling of the Moments

We now generalize the results obtained in the previous section, designing a complete scaling scheme for the moments of order *m* in the three time regimes, when different advection field conditions are analyzed.

Generally speaking, when the phase disorder is quenched, and for M→∞, the overall majority of the systems will display apparent scaling behavior only for even moments, while the odd moments show large sample-to-sample fluctuations around zero. Thus, for a generic quenched disorder, the even moments averaged over the initial conditions display a *typical* scaling, i.e., compatible with the scaling of moments averaged over a statistical ensemble of the phases configurations (average over the disorder). The cases where the disorder is absent, i.e., when $\phi_{n}=0$∀n, are not typical. They are indeed characterized by moment scalings in time different from those obtained from the moments averaged over the disorder. Moreover, while the asymmetric case of sine advection field (Equation 3) does not possess well-defined odd moments scaling, this can be evaluated explicitly in the case of symmetric shear velocity field (Equation 4).

### 4.1. Disorder-Average and Random Phase Configuration

A direct, although tedious, calculation shows that the scaling of the moments averaged over the disorder is
(34)$${\langle {\langle {\left[x(t)-x(0)\right]}^{m}\rangle }_{w}\rangle }_{\phi }∼\left\{\begin{array}{cc} L^{\frac{m}{2}}t^{m} & t\ll \frac{L^{2}}{M^{2}D}\\ L^{\frac{m}{2}}t^{m\frac{\left(3-\gamma \right)}{4}} & \frac{L^{2}}{M^{2}D}\ll t\ll \frac{L^{2}}{D}\\ L^{\frac{m}{2}}t^{\frac{m}{2}} & t\gg \frac{L^{2}}{D}. \end{array}\right.$$

Here, the index *m* can assume only even values, as the odd moments are identically zero. The expression (Equation 34) is confirmed by the numerical simulations in Figure 5 (solid black lines), where the moments show a perfect agreement with the theoretical estimation. Moreover, rescaling the moments shown in panel (b), ${\langle {\langle {\left[x(t)-x(0)\right]}^{m}\rangle }_{w}\rangle }_{\phi }^{1/m}$ yields a perfect collapse of the different moments in each of the three regimes. This is a bright indication that the even moments exhibit an anomalous scaling [29,30,31] in the three regimes.

We have seen in the Section 3.2 and Section 3.3 (Figure 4) that the third and fourth moments, averaged over the initial conditions, exhibit the average trend of ${\langle {\langle \left[{x\left(t\right)-x\left(0\right)]}^{m}\right]\rangle }_{w}\rangle }_{\phi }$. However, for a generic choice of the phases, fluctuations around this mean appear to increase with time. While these sample-to-sample fluctuations are considerably marked in case of odd moments, wildly scattered around zero, when *m* is an even number they do not appreciably affect the scaling in (Equation 34). This is indeed shown in Figure 5, where a single realization of ${\langle {\langle \left[{x\left(t\right)-x\left(0\right)]}^{m}\right]\rangle }_{w}\rangle }_{0}$ obtained for a random choice of $\left\{\phi_{n}\right\}$ (colored symbols) overlaps with the disorder average moments (Equation 34).

### 4.2. No-Disorder: Symmetric and Asymmetric Quenched Velocity Fields

When the no-disorder condition is enforced, i.e., $\phi_{n}=0\;\forall n$, the scaling of the moments differs substantially from the disorder-averaged case (Equation 34). A lengthy calculation, although conceptually non-demanding and equivalent to those reported in Appendix A, shows that, when the symmetric velocity advection field (Equation 4) is chosen, even and odd moments exhibit clear scaling forms in time (see solid lines in Figure 6). In particular, it appears that these scaling expressions have a continuous linear dependence on *m* in the range of short to intermediate times $t\ll \frac{L^{2}}{D}$, i.e., $\propto L^{m-1}t^{m}$ for $t\ll \frac{L^{2}}{M^{2}D}$ and $\propto L^{m-1}t^{m\frac{\left(2-\gamma \right)}{4}+\frac{1}{2}}$ for $\frac{L^{2}}{M^{2}D}\ll t\ll \frac{L^{2}}{D}$. However, asymptotically they behave discontinuously like $∼L^{m-1}t^{\left[m/2\right]}$, where [⋯] represents the integer part. As a matter of fact, a non-linear rescaling of the curves like that displayed in panel (b) of Figure 6, i.e., ${\left(\frac{{\langle {\langle {\left[x(t)-x(0)\right]}^{m}\rangle }_{w}\rangle }_{0}}{\sqrt{t}}\right)}^{1/m}$ works fine in only in the time range $L^{m-1}t^{m\frac{\left(2-\gamma \right)}{4}+\frac{1}{2}}$. This finding sheds light on the strong anomalous character of the spectrum of moments [29,30,31].

When the velocity field is asymmetric, i.e., (Equation 3), the odd moments present large fluctuations in time, although they are zero on average. Therefore, a precise scaling form in time cannot be inferred. The even moments instead follow the same scaling form of the symmetric case in the short to intermediate time regime $t\ll \frac{L^{2}}{D}$, and they coincide for large times (dashed lines in Figure 6).

Summing up, the results for the symmetric and asymmetric advection fields are hereby reported
(35)$${\langle {\langle {\left[x(t)-x(0)\right]}^{m}\rangle }_{w}\rangle }_{0}∼\left\{\begin{array}{cc} L^{m-1}t^{m} & t\ll \frac{L^{2}}{M^{2}D}\\ L^{m-1}t^{m\frac{\left(2-\gamma \right)}{4}+\frac{1}{2}} & \frac{L^{2}}{M^{2}D}\ll t\ll \frac{L^{2}}{D}\\ L^{m-1}t^{\left[\frac{m}{2}\right]} & t\gg \frac{L^{2}}{D}. \end{array}\right.$$

The former Equation (Equation 35) holds for any *m* when the velocity field is symmetric, namely in the cosine case (Equation 4). Its validity is limited to only values of 2m for antisymmetric velocity fields (Equation 3).

## 5. Conclusions

We have furnished a thorough analysis of the random shear model in the case of long-ranged advection fields. The strength of the correlations along *y* is dictated by the exponent of the discrete spectrum of the velocity field $∼k_{n}^{\gamma }$. The elegant way in which the random shear velocity is introduced presents several advantages if compared to the previous analysis, among which we can enumerate the possibility to derive exactly all the advection (shear) field statistical properties by means of the n−points correlation functions and of the PDF P(U). Importantly, it allows the straightforward implementation of two kinds of averaging procedures, one ${\langle {\langle \cdots \rangle }_{w}\rangle }_{\phi }$ over the thermal noise *w* and the shear field disorder embodied by the phases $\left\{\phi_{n}\right\}$, the other ${\langle {\langle \cdots \rangle }_{w}\rangle }_{0}$ over the thermal noise *w* and a uniform set of initial conditions $y_{0}$ spaced along the channel’s size *L*, for a quenched configuration of the disorder. In particular, we focused on the comparison between the scaling form in time of the position moments achieved by averaging over the disorder ensemble, and the scaling exhibited by the moments averaged over the initial conditions. Three distinct cases of the quenched advection field were thoroughly analyzed: *i*) a typical random configuration of the phases $\left\{\phi_{n}\right\}$, *ii*) the case of the asymmetric advection field (Equation 3) with no disorder $\phi_{n}=0\;\forall n$, iii) the case of the symmetric advection field (Equation 4) with no disorder $\phi_{n}=0\;\forall n$.

The four cases analyzed show the presence of three distinct scaling regimes for $t\ll \frac{L^{2}}{M^{2}D}$, $\frac{L^{2}}{M^{2}D}\ll t\ll \frac{L^{2}}{D}$ and $t\gg \frac{L^{2}}{D}$. In the case of average over the disorder, the moments ${\langle {\langle {\left[x(t)-x(0)\right]}^{m}\rangle }_{w}\rangle }_{\phi }$ display a typical scaling form of the type $\propto t^{\alpha m}$ in the three regimes, only for even values of *m*. On the other hand, they are equal to 0 for odd *m*. The scaling of the moments ${\langle {\langle {\left[x(t)-x(0)\right]}^{m}\rangle }_{w}\rangle }_{0}$ when a generic random configuration of the phases $\left\{\phi_{n}\right\}$ has been set up, traces that of ${\langle {\langle \left[{x\left(t\right)-x\left(0\right)]}^{m}\right]\rangle }_{w}\rangle }_{\phi }$ only when *m* is even. The odd moments show random sample-to-sample fluctuations which, obviously, are not consistent with any scaling law. In this sense, we can state that the average over the initial conditions for a generic quenched disorder is typical, and follows a universal scaling. The cases of zero disorder exhibit scaling trends differ from those of a generic random configuration for times $t\gg \frac{L^{2}}{M^{2}D}$. Specifically, the symmetric advection field (Equation 4) has moments whose scaling form is strongly anomalous for any *m*. In the antisymmetric case (Equation 3), the even moments behave exactly like the symmetric counterparts. The odd moments, on the other side, are random functions of time for which a clear scaling form cannot be drawn.

Finally, in this article, we offered the most detailed study of the scaling form of the position moments in the random shear model. Our findings include and extend to all the previous results present in the literature, providing a comprehensive and accurate theoretical framework on this subject. Moreover, it constitutes a benchmark for future analysis including the analytical calculations of the propagator P(x,t), the diffusion equation that it satisfies and the question of the ergodicity breaking in either one of the two averaging procedure.

## Figures and Tables

**Figure 1 entropy-24-01350-f001:**
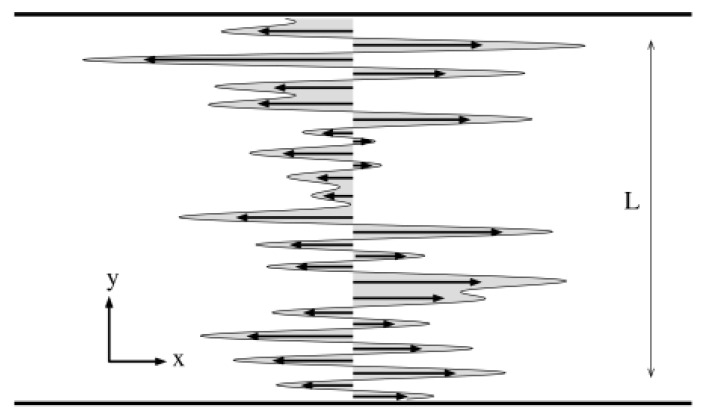
Random shear flow schematic view. The channel, where the tracer’s stochastic motion takes place, has width *L* in the transverse direction *y* and it is assumed infinite along the longitudinal coordinate *x* (in the numerical simulations periodic boundary conditions are enforced). The quenched disorder velocity field is depicted as black arrows pointing only along *x*, the grey envelope corresponds to the superposition of sinusoidal waves according to the definitions (Equation 3) and (Equation 4).

**Figure 2 entropy-24-01350-f002:**
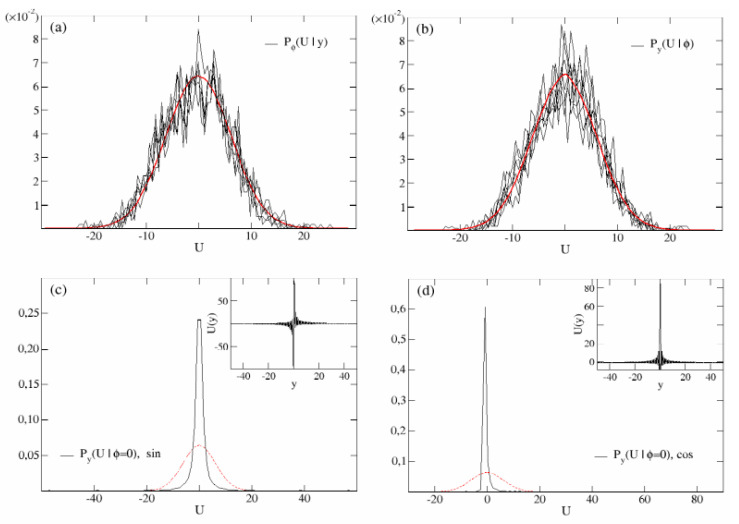
Shear flow PDF. Panel (**a**): the black curves represent the shear field PDF $P_{\phi }\left(U|y\right)$ of the velocity U(y) collected at 10 different positions *y*. The values of *U* in each curve have been obtained by simulating an ensemble of 100 different configurations of the phases $\left\{\phi_{n}\right\}$. Simulations parameters are γ=−0.4, L=100 and M=100. Being actually independent of *y*, the average Gaussian character of $P_{\phi }\left(U|y\right)$ is apparent by increasing the size of the disorder statistical ensemble (red curve, $10^{4}$ independent configurations $\left\{\phi_{n}\right\}$). Panel (**b**): shear flow PDF $P_{y}\left(U|\phi \right)$ obtained from the values of U(y) collected along the channel y∈[−50,50], for several 10 independent quenched configurations of the phases $\left\{\phi_{n}\right\}$ (black curves). The collected values of U(y) involve 2000 uniformly distributed points along the *y* axis. The red curve represents the same PDF for the velocity field U(y) sampled over $10^{5}$ points along the channel’s width. The other simulation parameters coincide with those of panel (a). Panel (**c**): same quantity as in panel (**b**) obtained from the velocity field in the inset, namely with U(y) given by (Equation 3) and $\phi_{n}=0,\forall n$. The shear field in the inset shows the presence of antisymmetric long jets close to y=0 and, smaller values elsewhere. Other simulation parameters coincide with those of panel (**a**,**b**). The red curve represent the Gaussian P(U) shown in panels (**a**,**b**). Panel (**d**): same quantity as in panel (**b**,**c**) obtained from the velocity field in the inset, namely with U(y) given by (Equation 4) and $\phi_{n}=0,\forall n$. In the inset the shear field is dominated by symmetric shoots close to the origin (U(−y)=U(y)) surrounded by negative values of the velocity U(y). The red curve represent the Gaussian P(U) shown in panels (**a**,**b**). Other simulation parameters coincide with those of panel (**a**–**c**).

**Figure 3 entropy-24-01350-f003:**
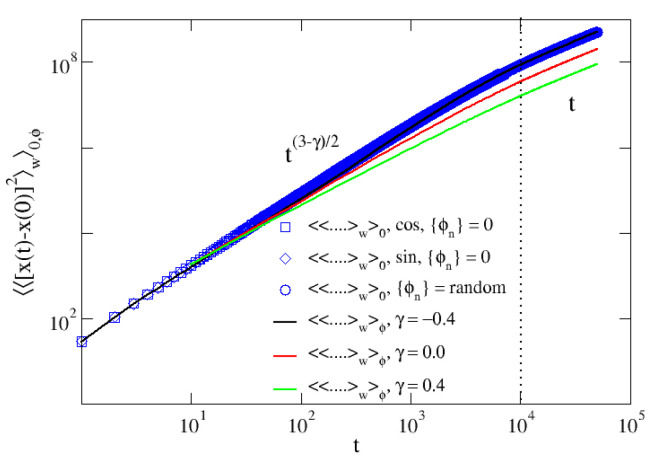
Second moment. Mean square displacement averaged over the disorder (solid lines) and over the initial conditions (blue symbols). The scaling regimes in (Equation 27) are apparent: the dotted line stands for $t=\frac{L^{2}}{D}$ with L=100D=1.0 and M=100. The averages over the initial conditions are performed over $10^{4}$$y_{0}$ equally spaced along the channel’s width *L*.

**Figure 4 entropy-24-01350-f004:**
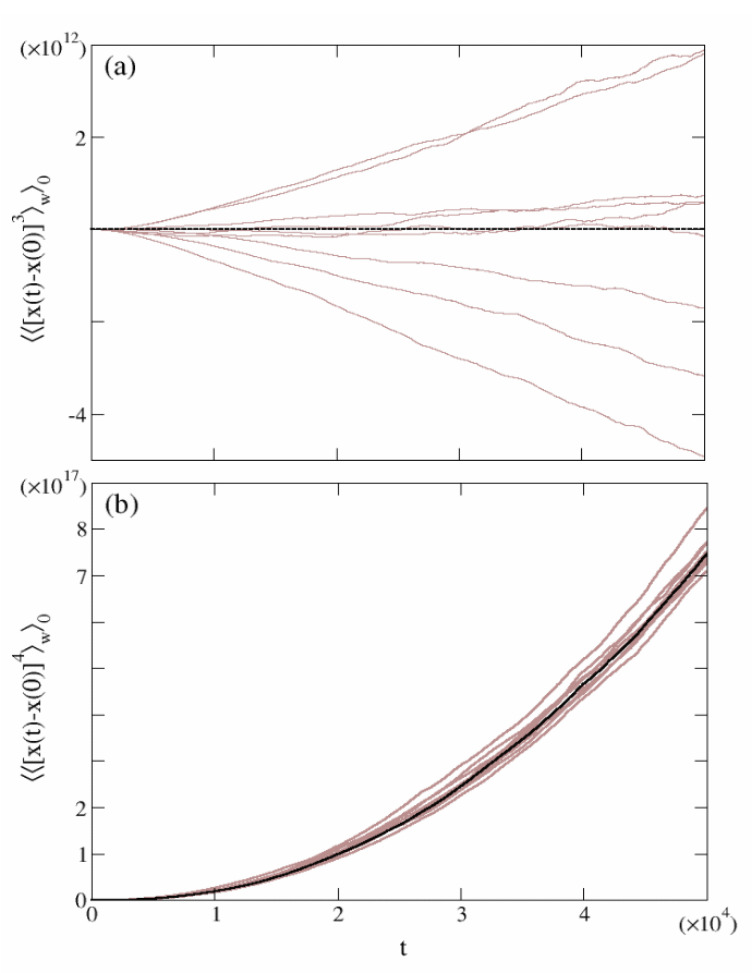
Sample-to-sample fluctuations. The brown solid line represents the time behavior of the third (panel (**a**)) and fourth moments (panel (**b**)) of 10 different random configurations of the disorder, averaged over $10^{4}$ initial positions. The dashed lines stand for the disorder-averaged ${\langle {\langle \cdots \rangle }_{w}\rangle }_{\phi }$. The other simulations parameters are L=100, γ=−0.4, D=1.0 and M=100.

**Figure 5 entropy-24-01350-f005:**
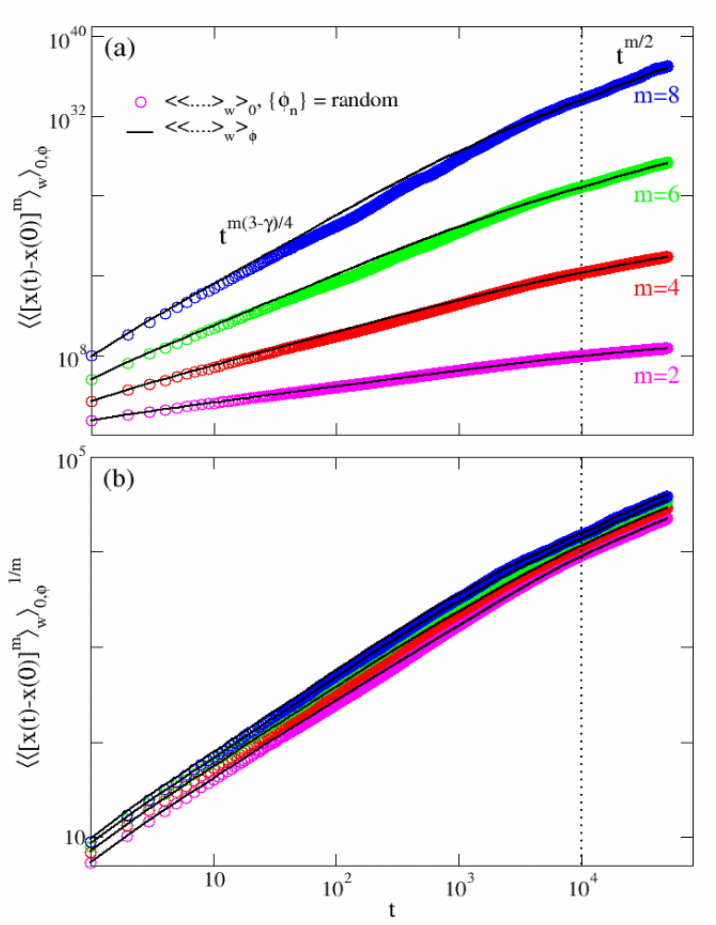
Scaling of moments: disordered case. Panel (**a**): scaling regimes for the moments averaged over 10 configurations of the disorder (black solid lines) and over $10^{4}$ initial conditions for a quenched random configuration of the disorder. The dotted vertical line correspond to $t=\frac{L^{2}}{D}$ with L=100, D=1.0 and M=100. The expression (Equation 34) is supported by the rescaling in panel (**b**).

**Figure 6 entropy-24-01350-f006:**
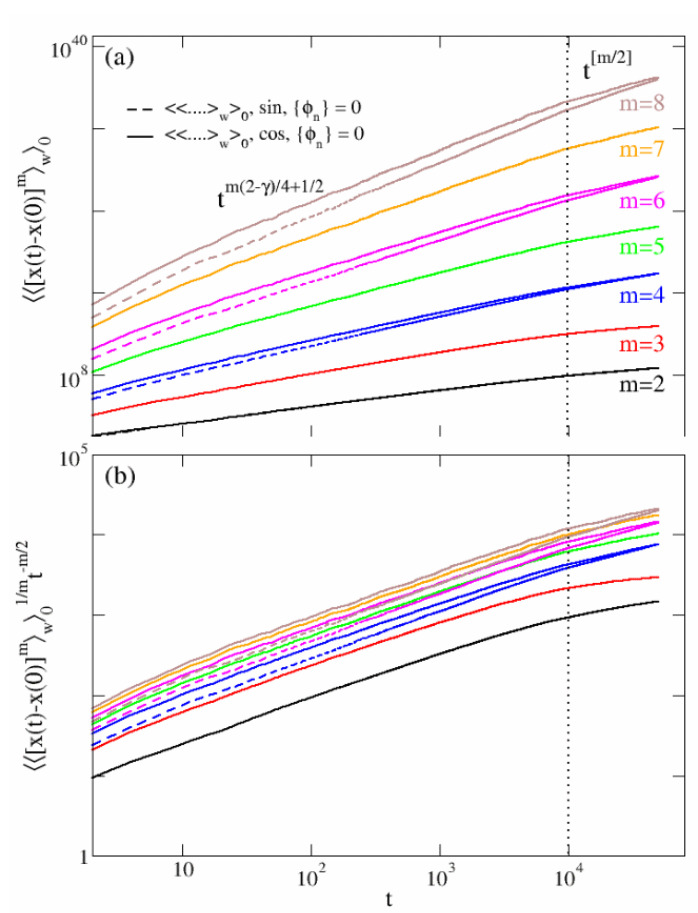
Scaling of moments: no-disorder case. Panel (**a**): scaling regimes for the moments averaged over $10^{4}$ initial conditions for $\{\phi_{n}\}=0$. The solid lines correspond to the symmetric case (Equation 4), while the dashed lines stand for the asymmetric case (Equation 3). The dotted vertical line correspond to $t=\frac{L^{2}}{D}$ with L=100, D=1.0 and M=100. The rescaling in panel (**b**) show a perfect curves collapse in the regime $\frac{L^{2}}{M^{2}D}\ll t\ll \frac{L^{2}}{D}$ but it does not hold for $t\gg \frac{L^{2}}{D}$.

## Data Availability

The code generating the data are reported in https://github.com/Nevi-code/random-shear/ (accessed on 4 August 2022).

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
