# Peer review of "Universal and Non-Universal Features in the Random Shear Model"

_entropy, 2022, doi:10.3390/e24101350_

Round 1

Reviewer 1 Report

The authors study stochastic properties of the two-dimensional
so-called random shear model, where random (quenched) amplitudes
layered along the y axis are sampled randomly according to overdamped
Langevin-type dynamics, thus modeling both advection and diffusion. In
particular, they are interested in the impact of different types of
averaging (over thermal noise, phases and initial conditions) on the
calculation of higher-order correlation functions and moments
characterising the emerging stochastic dynamics. They explore these
properties for three different types of advection fields and find
especially particular scaling forms for the moments as functions of
time.

The ms.\ is altogether well written, and the results appear to be
quite strong (as far as I can judge, since I am not an expert in this
field). I have no major comments, only a number of rather minor
remarks that I may ask the authors to take into account. After
suitable modifications, in my view this ms.\ will form a very worthy
contribution to Entropy.\\

1. This remark may be a bit off, but the random shear model reminds me
to some extent to what has been studied as the `comb model' in
stochastic theory (see, e.g., work by Alexander Iomin and many
others). Just wanted to bring this to the authors' attention, though
probably in detail there is no closer relation here?

2. I think the caption of Fig.1 should be a bit more detailed.

3. There is a typo on p.3 l88, the cited eqs.\ should be (3), (4), not (6), (7).

4. Another typo on p.4, l110, Fig.1(a) should read 2(a).

5. p.9, Eq.(28)ff: I am curious about the transition points between
the scalings in the three different temporal regimes. For transport in
anomalous stochastic models like continuous time random walks or
generalised Langevin equations it is known that transitions between
different types of diffusive dynamics are marked by multiplicative
logarithmic terms in time that survive in the long time limit right at
the transition points but decay on shorter times scales around these
transition points (for CTRWs see, e.g., standard literature on Levy
walks or subdiffusive models). Is it conceivable that something like
this happens here as well? Because otherwise the time dependence of
the moments would be continuous but non-differentiable while the
simulation results show rather smooth curves.

6. p.13, Sec.3.2: I got a bit lost here in view of the presentation of
the results. First the time scaling for the symmetric velocity field
is given in the text (without prefactors). Then the asymmetric field
is discussed leading to Eq.(36) (with prefactors)? Or is this
eq.\ supposed to `summarise' both cases (as the time scaling appears
to be the same)? But for the asymmetric field the odd moments are zero
while in (36) there is no constraint on $m$?  Please check/clarify.

7. There are a few typos in the text, and some citations/references
are not resolved in the Supplement yielding question marks.

Reviewer 2 Report

Manuscript title: Universal and non-universal features in the random shear model

Authors: Fabio Cecconi, Alessandro Taloni

Comments:

The authors have analyzed the universality (or lack thereof) of a random shear model, and the appropriate conditions have been identified. I recommend the article for publication, but I would like the authors to address the following comments in their revised manuscript:

1.      The authors could improve their description of the system. Figure 1 and the accompanying text are not very clear.

2.      The connection between the stochastic processes studied and the actual physics is lacking or not clear. For instance, is it possible to observe random or quenched disorder in practice or which one is more probable to encounter. It would be good to have a paragraph or two describing this connection.

3.      The significance of higher-order moments is not very clear in the analysis presented. It would be beneficial to the readers if the authors could describe the need for moments higher than order 2 to describe a system.

4.      It would add value if the authors could discuss ergodicity in these processes. My guess is with the measurements already made; it would be straightforward to prove weak ergodicity breaking in these systems. Refer to the work of Metzler et al. (Metzler, R., Jeon, J. H., Cherstvy, A. G., & Barkai, E. (2014). Anomalous diffusion models and their properties: non-stationarity, non-ergodicity, and ageing at the centenary of single particle tracking. Physical Chemistry Chemical Physics16(44), 24128-24164.).

5.      The anomalous diffusion process described by the mean-squared displacement is interesting, in particular, the cross-over from super-diffusion at short and intermediate times to normal diffusion for longer durations (Equation 28). One possible explanation is that the occurrence of trap events increases with time, thus slowing down the overall diffusion process. Similar observations have been made in physical systems, here are a couple of examples:

1.      Tabei, S. A., Burov, S., Kim, H. Y., Kuznetsov, A., Huynh, T., Jureller, J., ... & Scherer, N. F. (2013). Intracellular transport of insulin granules is a subordinated random walk. Proceedings of the National Academy of Sciences110(13), 4911-4916.

2.      Higham, J. E., Shahnam, M., & Vaidheeswaran, A. (2021). Anomalous diffusion in a bench-scale pulsed fluidized bed. Physical Review E103(4), 043103.

Discussing these results in the context of the work presented would add a new perspective, and possibly aid future research.

6.      Figure 2 in Page 5 is not clear. The plots and font sizes could be enlarged.

7.      The paper is well-written for the most part, but not free from grammatical or typographical errors. I urge the authors to have a thorough proof-reading done before submitting the revised version.
